# Process Optimization for Compression Molding of Carbon Fiber–Reinforced Thermosetting Polymer

**DOI:** 10.3390/ma12152430

**Published:** 2019-07-30

**Authors:** Jiuming Xie, Shiyu Wang, Zhongbao Cui, Jin Wu

**Affiliations:** 1School of Mechanical Engineering, Tianjin University; Tianjin 300072, China; 2Tianjin Sinotech Industry Co., Ltd., Tianjin 301700, China; 3School of Mechanical Engineering, Tianjin Sino-German University of Applied Science; Tianjin 300350, China

**Keywords:** PAN-based CFRTP, compression molding, test, optimization of process parameters

## Abstract

To enhance the quality and mechanical performance of a carbon fiber–reinforced polymer (CFRP) workpiece, this paper prepares a polyacrylonitrile (PAN)-based carbon fiber–reinforced thermosetting polymer (CFRTP) laminated board through compression molding, and carries out orthogonal tests and single-factor tests to disclose the effects of different process parameters (i.e., compression temperature, compression pressure, pressure-holding time, and cooling rate) on the mechanical performance of the CFRTP workpieces. Moreover, the process parameters of compression molding were optimized based on the test results. The research results show that: The process parameters of compression molding can be ranked as compression temperature, pressure-holding time, compression pressure, cooling rate, and mold-opening temperature, in descending order of the impact on the mechanical property of the CFRTP; the optimal process parameters for compression molding include a compression temperature of 150 °C, a pressure-holding time of 20 min, a compression pressure of 50 T, a cooling rate of 3.5 °C/min, and a mold-opening temperature of 80 °C. Under this parameter combination, the tensile strength, bending strength, and the interlaminar shear strength (ILSS) of the samples were, respectively, 785.28, 680.36, and 66.15 MPa.

## 1. Introduction

The carbon fiber–reinforced polymer (CFRP) is widely known for its excellence in specific strength, specific modulus, fatigue resistance, corrosion resistance, forming process, damage safety, and functional designability [1,2,3]. Since the turn of the century, the CFTP has been extensively applied in aerospace engineering, wind turbine blades, sports equipment, and auto parts, becoming the dominant material of lightweight products. More than 90% of the CFRP market is occupied by carbon fiber–reinforced thermosetting polymer (CFRTP), which has enjoyed rapid development thanks to its simple fabrication, fast processing, and high cost performance [4,5].

The CFRTP can be molded through compression molding, autoclaving, winding, or pultrusion [6,7,8]. Compression molding stands out for its low cost, high efficiency, low internal stress, small buckling deformation, good mechanical stability, and excellent product repeatability [9]. Unsurprisingly, this molding method boasts a strong competitive advantage in industrial mass production of parts and components, and the advantage grows with the production volume. However, the process parameters of compression molding (e.g., preheating temperature, molding temperature, molding pressure, pressure holding time, cooling rate, exhaust pressure, exhaust times, and blank holder force) directly affect the flow of the matrix material and the impregnation effect of the reinforcing fibers [10,11,12,13]. This effect, coupled with the interaction between process parameters, exerts an impact on the quality and mechanical performance of the workpiece. To optimize the mechanical performance of the workpiece, it is critical to analyze the interaction between various factors and determine the best process parameters of compression molding [14,15,16].

In order to optimize the process parameters of compression molding of CFRTP, a lot of research and exploration have done around the establishment of mathematical models, finite element analysis, and experimental verification [17,18,19,20,21,22,23,24]. In the field of experimental verification especially, fruitful results have been achieved. By designing the orthogonal test, the sample space is reduced effectively and the test cost is reduced. The mapping relationship between molding process parameters and molding quality was obtained by single-factor tests, which laid a foundation for the study of the mechanism of compression molding. In this paper, the continuous carbon fiber (CCF300)–reinforced polyacrylonitrile (PAN) resin composite was taken as the object, and five process parameters were selected to characterize the compression molding, namely compression temperature, compression pressure, pressure holding time, cooling rate, and mold-opening temperature. Orthogonal tests and single-factor tests were designed to analyze the tensile strength, bending strength, and interlaminar shear strength (ILSS) of workpieces made through compression molding. In light of the test results, the author discussed how each process parameter affects the mechanical performance of the CCF300-reinforced PAN resin composite laminated board, and then optimized the process parameters of compression molding. The research findings shed new light on process optimization, quality control, and molding mechanism of the CFRTP.

## 2. Materials and Methods 

### 2.1. Raw Materials

The CFRTP is the prepreg supplied from Shanghai TRONXT New Material Technology Co., Ltd. (Shanghai, China), with the carbon fiber as long fiber of Toray T300 and PAN-based carbon fiber as the matrix. The characteristic parameters of the prepreg are listed in Table 1.

The prepreg is shown in Figure 1.

### 2.2. Test Equipment

#### 2.2.1. Equipment for Compression Molding

The compression molding was conducted on a 2000 kN intelligent mechanical servo universal testing machine, custom-made by Tianjin Tianduan Press Co., Ltd (Tianjin, China). The machine (nominal force: 2000 kN; slider stroke: 600 mm; bed dimension: 1800 mm × 1300 mm) can simulate the molding process curves of many machine tools, such as crank press and hydraulic press, and collect real-time process parameters in equipment movement. In addition, the machine is supported by a handling robot system, a rapid cooling and heating machine, a conveying system, and a forming mold. The forming mold is a custom-made carbon fiber flat forming mold with a forming size of 500 mm × 500 mm. The equipment for compression molding is shown in Figure 2.

#### 2.2.2. Equipment for Sample Cutting

The samples were cut by a four-axis cantilever waterjet cutting machine (Shenyang HEAD Science & Technology Co., Ltd., Shenyang, China). Under the control of the computer, the machine (effective stroke: 1500 mm × 2000 mm × 150 mm) can cut the workpiece into any form and also trim the formed composite parts to the required size, without being greatly affected by the material texture. The equipment for sample cutting is shown in Figure 3.

#### 2.2.3. Equipment for Testing 

The tests were carried out on an universal material testing machine (MTS Systems Corp, Minneapolis, America). The machine (maximum test force: 50 N~10 kN; effective space of tension: 700 mm; effective test width: 300 mm) is mainly used for testing and analysis of tensile, compressive and bending properties of nonmetallic materials. With three closed-loop control modes (i.e., stress, strain, and displacement), the machine helps to determine such parameters as the maximum force, tensile strength, bending strength, compressive strength, and elongation at break. The equipment for testing is shown in Figure 4.

### 2.3. Sample Preparation

To prepare the samples, the carbon fiber prepreg was laid down and covered with a piece of release cloth on the surface to prevent adhesion after heating. Next, the prepreg was relocated to a drying oven for heating. The temperature was maintained at the pre-set value. Finally, the prepreg was taken out and quickly transferred to the mold for compression molding. The entire process was implemented on the automatic assembly line.

The compression molding process of our PAN-based CFRTP is illustrated in Figure 5 below.

The prepreg was processed in the following steps: First, cut the carbon fiber prepreg into 495 mm × 495 mm to suit the mold size; Then, place the prepreg into an oven to dry for 4 h at 45 °C under near-vacuum conditions; after that, lay the prepreg fibers interlaced at 90° within the flat mold, followed by preheating, heating, pressurizing, pressure holding, cooling, and unloading; finally, the sample was demolded and cut into standard test samples by the four-axis cantilever waterjet cutting machine. The carbon fiber broad after compression molding is shown in Figure 6 (non-optimal process parameters).

### 2.4. Characterization and Testing

According to the ASTM D3039 test standard [25], each tensile test sample was prepared into a dumbbell-shape with a width of 25 mm and a total length of 250 mm and subjected to tensile loading at 2 mm/min with the standard strain rate of 0.01 min^−1^.

According to the ASTM D790 test standard [26], each bending test sample was prepared with a span thickness ratio of 32:1 and a width of 13 mm, prestressed to 5 N, and subjected to bending at 1 mm/min with the test length exceeding the span by 20%.

According to the ASTM D2344 test standard [27], each ILSS test sample was prepared with a width of 25 mm and a length of 250 mm, and subjected to shearing at 2 mm/min.

The sample after cutting based on ASTM D2344 is shown in Figure 7.

### 2.5. Test Orthogonal Plan

The properties of PAN-based CFRTC are influenced by many process conditions of compression molding. This paper mainly targets five process parameters, namely, compression temperature, pressure holding time, compression pressure, cooling rate, and mold-opening temperature. Table 2 shows the levels of the five parameters in the orthogonal tests.

The orthogonal tests mainly investigate the effects of the five parameters on the mechanical performance of the CFRTP made through compression molding, without considering the interaction between the parameters. The L_16_(4^5^) orthogonal scheduling (Table 3) was adopted for the tests [28,29].

## 3. Results and Discussion of Orthogonal Testing

### 3.1. Result of Orthogonal Test

The standard test samples were prepared according to Table 3, and subjected to tensile strength, bending strength, and ILSS tests on the universal material testing machine. In order to test the results accurately, each molded laminate was randomly cut into 20 samples, then tensile strength, bending strength, and ILSS were tested in turn, the final measurement results were averaged. The measurement error is indicated by the standard deviation.

The test results are recorded in Table 4.

### 3.2. Discussion of Orthogonal Test

The test results on the tensile strength of factor A are cited as an example to interpret the results of the orthogonal tests. As shown in Table 3, the first level of factor A, denoted as A_1_, exerted an impact on the tensile strengths measured in tests 1–4, the second level of factor A, denoted as A_2_, exerted an impact on the tensile strengths measured in tests 5–8, the third level of factor A, denoted as A_3_, exerted an impact on the tensile strengths measured in tests 9–12, and the fourth level of factor A, denoted as A_4_, exerted an impact on the tensile strengths measured in tests 13–16.

The sum of all tensile strengths under A_1_, denoted as $\delta_{A_{1}}$, can be expressed as:(1)$$\delta_{A_{1}}=742.16+753.45+712.26+724.13=2932.00\;\mathrm{MPa}$$

The mean of all tensile strengths under A_1_, denoted as $\bar{\delta_{A_{1}}}$, can be expressed as:(2)$$\bar{\delta_{A_{1}}}=\frac{2932.00}{4}=733.00\;\mathrm{MPa}$$

The tensile strengths under A_2–4_ and those under factors B–D were obtained similarly and recorded in Table 5.

Different factors led to test results in varied ranges. In general, a large range means the corresponding factor has a high impact on the test results. Thus, the factor leading to the largest range must be the dominant factor [30]. The range R can be calculated by:(3)$$R=\max\left(\bar{\delta_{k}}\right)-\min\left(\bar{\delta_{k}}\right)\;(k\;=\;1,2,3,4)$$
where $\max\left(\bar{\delta_{k}}\right)$ and $\min\left(\bar{\delta_{k}}\right)$ are the maximum and minimum of the arithmetic mean of the test results under level k of any factor, respectively.

The test design determines that the test conditions for A_1_, A_2_, A_3_, and A_4_ have exactly the same comprehensive comparability. As a result, the ranges of the tensile strength, bending strength, and the ILSS in the orthogonal tests could be computed separately. The calculated results are listed in Table 6 below.

As shown in Table 6, factors A and E had greater impacts than the other three factors on tensile strength, bending strength, and the ILSS, while factor B showed relatively small impacts on the three mechanical performance. Meanwhile, the process parameters differed slightly in their effects on bending strength and the ILSS, but significantly in their effects on tensile strength. Overall, the five parameters could be ranked as compression temperature, pressure-holding time, compression pressure, cooling rate, and mold-opening temperature, in descending order of the effect on mechanical performance of the workpieces.

## 4. Results and Discussion of Single-Factor Tests 

According to the degree of impacts of each process parameter on mechanical performance (i.e., tensile strength, bending strength, and the ILSS), the operation parameters were increased and adjusted for single-factor tests on compression temperature, pressure-holding time, compression pressure, and cooling rate, respectively. During the tests, one of the process parameters is changed, and the other process parameters are unchanged. Through the single-factor tests, the mapping relationship between the parameters and the mechanical properties as same as the influencing mechanism were analyzed. Since the ILSS is little affected by the process parameters of compression molding, the single-factor tests only consider the tensile strength and the bending strength.

### 4.1. Effects of Compression Temperature 

Table 7 presents how compression temperature affects the mechanical performance of the CFRTC samples made through compression molding at the pressure-holding time of 20 min, the compression pressure of 50 T, the cooling rate of 3.5 °C/min, and the mold-opening temperature of 80 °C.

Based on Table 7, the mapping relationship between compression temperature and mechanical properties is shown in Figure 8.

It can be seen from Figure 8 that the mechanical performance was optimized under the compression temperature of 150 °C and was not greatly affected when the compression temperature fell at 150–155 °C. These results can be explained as follows. In the molding process, if the heating temperature is too low, the resin cannot fully melt or flow, leading to high flow viscosity and insufficient impregnation; if the temperature is too high, the fibers will be ablated, and the resin will degrade, reducing the mechanical performance.

### 4.2. Effects of Pressure-Holding Time 

Table 8 presents how pressure-holding time affects the mechanical performance of the CFRTC samples made through compression molding at the compression temperature of 150 °C, the compression pressure of 50 T, the cooling rate of 3.5 °C/min, and the mold-opening temperature of 80 °C.

Based on Table 8, the mapping relationship between compression temperature and mechanical properties is shown in Figure 9.

As shown in Figure 9, the mechanical performance of the test material gradually increased with the elapse of the pressure-holding time, before the time reached a certain threshold. Any further growth in the time had a negligible effect on the mechanical performance. The mechanical performance remained constant after the threshold because the resin flow and impregnation both improve with the extension of the pressure-holding time, but the flow ceases after the resin is fully impregnated. Taking tensile strength as the main criterion, the optimal pressure-holding time was determined as 20 min for the mechanical properties of the samples.

### 4.3. Effects of Compression Pressure

Table 9 displays how compression pressure affects the mechanical performance of the CFRTC samples made through compression molding at the compression temperature of 150 °C, the pressure-holding time of 20 min, the cooling rate of 3.5 °C/min, and the mold-opening temperature of 80 °C.

Based on Table 9, the mapping relationship between compression pressure and mechanical properties is shown in Figure 10.

As can be seen from Figure 10, the mechanical performance of the samples improved and then declined with the continuous growth in compression pressure. A possible reason is that: The growing pressure provides a greater driving force to resin flow, thus increasing the impregnation rate and permeability of the resin. In this case, the porosity is reduced, and the PAN matrix and the fiber cross-section are bonded tighter than before. If the pressure is too high, the molten resin may overflow and damage the structure, resulting in a decrease in mechanical performance.

### 4.4. Effects of Cooling Rate

Table 10 showcases how cooling rate affects the mechanical performance of the CFRTC samples made through compression molding at the compression temperature of 150 °C, the pressure-holding time of 20 min, the compression pressure of 50 T, and the mold-opening temperature of 80 °C.

Based on Table 10, the mapping relationship between cooling rate and mechanical properties is shown in Figure 11.

Theoretically, the faster the cooling rate, the shorter the cooling time, the greater the residual stress inside the samples, the lower the geometric stability, and the more unstable the mechanical performance. In our test, however, the best mechanical performance was observed at the cooling rate of 3.5 °C/min, under the joint effect of multiple factors.

To sum up, our PAN-based CFRTP achieved the optimal mechanical performance under the following combination of process parameters: A compression temperature of 150 °C, the pressure-holding time of 20 min, a compression pressure of 50 T, a cooling rate of 3.5 °C/min, and a mold-opening temperature of 80 °C. Under this parameter combination, the tensile strength, bending strength, and the ILSS of the samples were respectively 785.28, 680.36, and 66.15 MPa. However, the optimum process parameters are the best parameters in the test state. Through the test, it is difficult to obtain the optimal process parameters of carbon fiber compression molding, and the interaction between theory and finite element analysis is needed.

## 5. Conclusions

The following conclusions were drawn from the study on the process parameters in the compression molding of PAN-based CFRTC: (i) During compression molding, the mechanical performance of the workpiece increased and then decreased with the growth in such process parameters as compression temperature, pressure-holding time, and compression pressure. Therefore, the mechanical performance of the CFRTC should be maximized by selecting the proper values of the process parameters for compression molding. (ii) During compression molding, the process parameters can be ranked as compression temperature, pressure-holding time, compression pressure, cooling rate, and mold-opening temperature in descending order of their impact on the tensile strength; as pressure-holding time, compression temperature, compression pressure, cooling rate, and mold-opening temperature in descending order of their impact on the bending strength; and as compression temperature, pressure-holding time, cooling rate, mold-opening temperature, and compression pressure in descending order of their impact on the ILSS. (iii) The PAN-based CFRTC achieved the optimal mechanical performance under the compression temperature of 150 °C, the pressure-holding time of 20 min, the compression pressure of 50 T, the cooling rate of 3.5 °C/min, and the mold-opening temperature of 80 °C. Under this parameter combination, the tensile strength, bending strength and the ILSS of the samples were respectively 785.28, 680.36, and 66.15 MPa. (iv) The optimal process curve was plotted according to the results of the orthogonal tests and the single-factor tests, laying a solid basis for intelligent forming and constitutive model.

## Figures and Tables

**Figure 1 materials-12-02430-f001:**
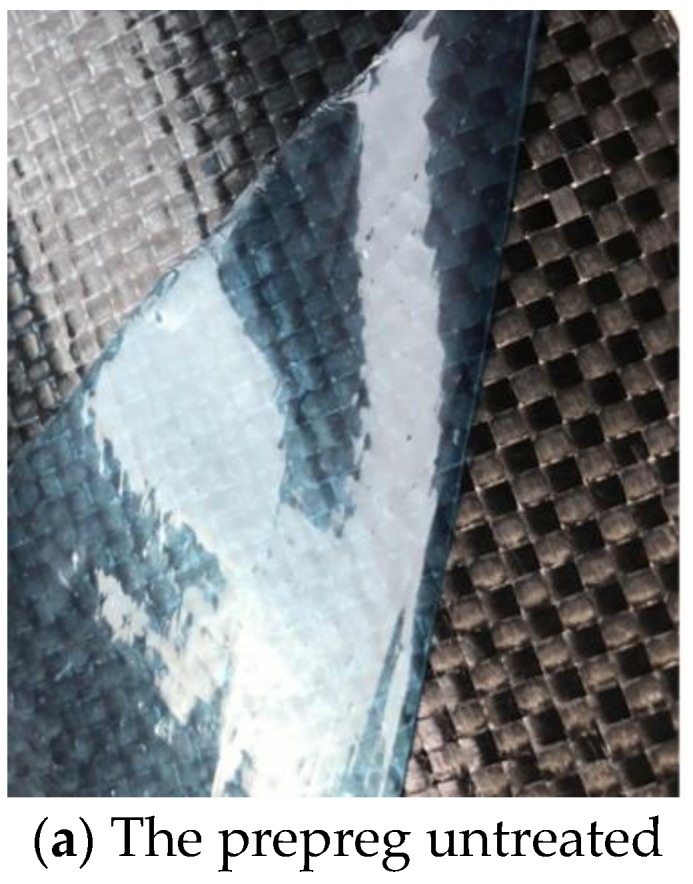
The prepreg.

**Figure 2 materials-12-02430-f002:**
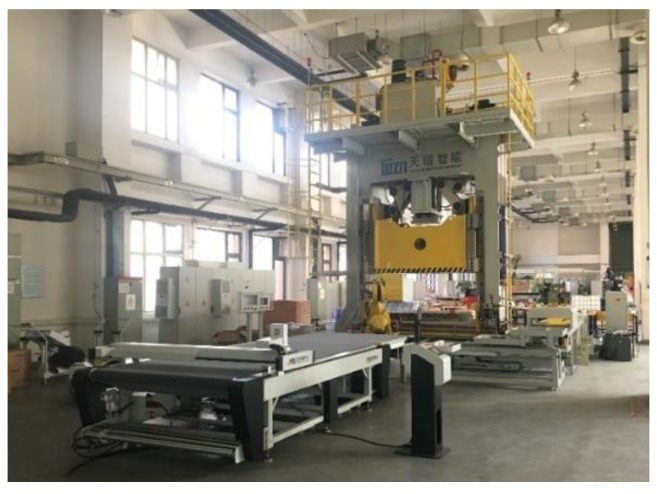
Equipment for compression molding.

**Figure 3 materials-12-02430-f003:**
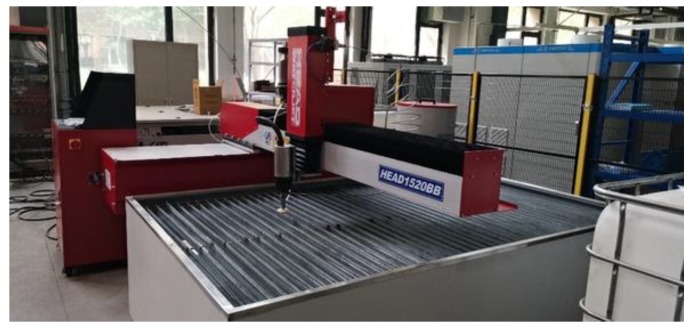
Equipment for sample cutting.

**Figure 4 materials-12-02430-f004:**
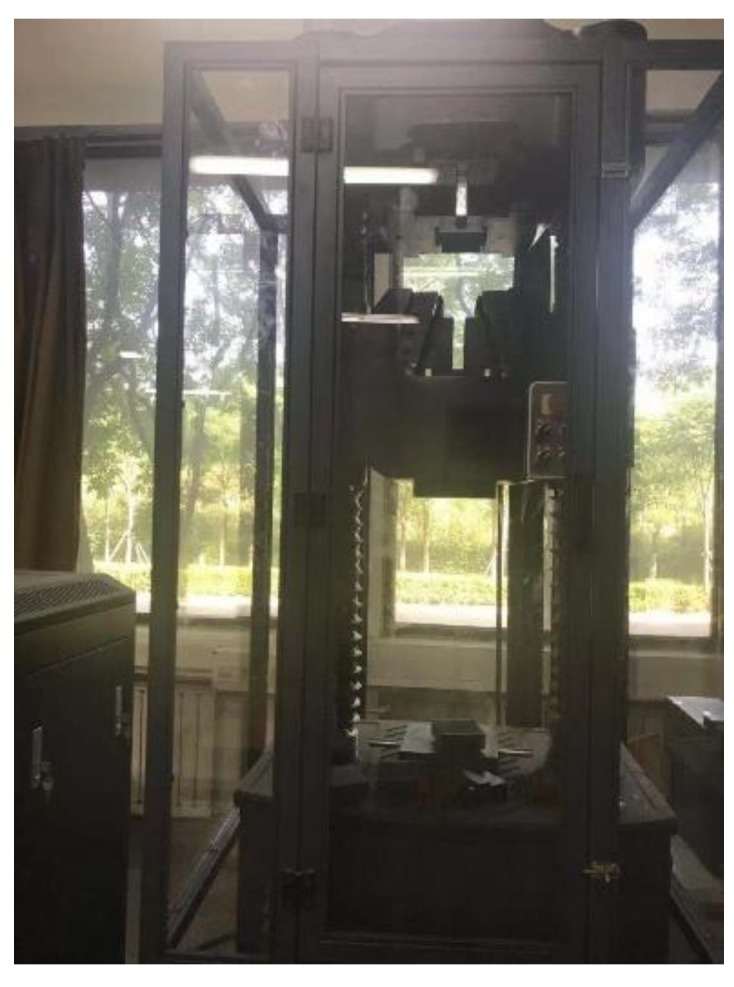
Equipment for testing.

**Figure 5 materials-12-02430-f005:**
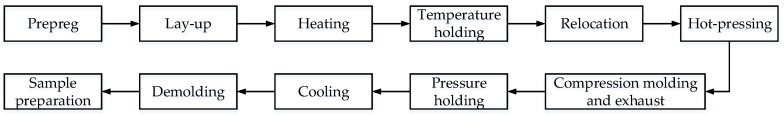
The compression molding process.

**Figure 6 materials-12-02430-f006:**
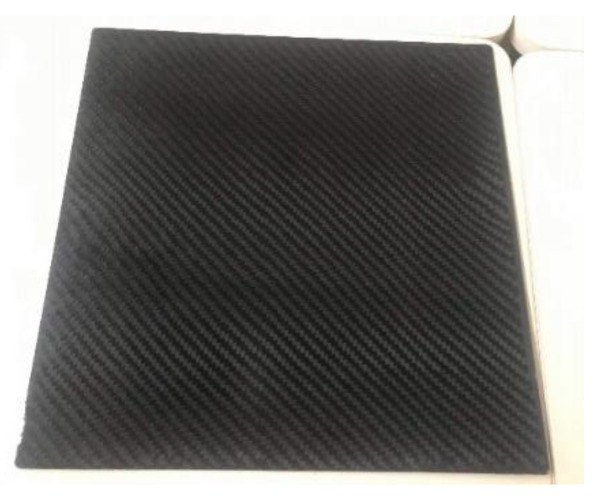
The carbon fiber broad after compression molding.

**Figure 7 materials-12-02430-f007:**
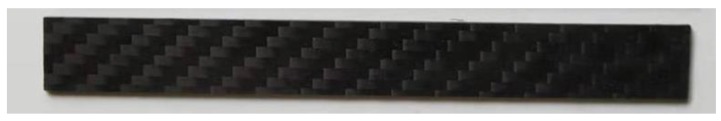
The sample after cutting based on ASTM D2344.

**Figure 8 materials-12-02430-f008:**
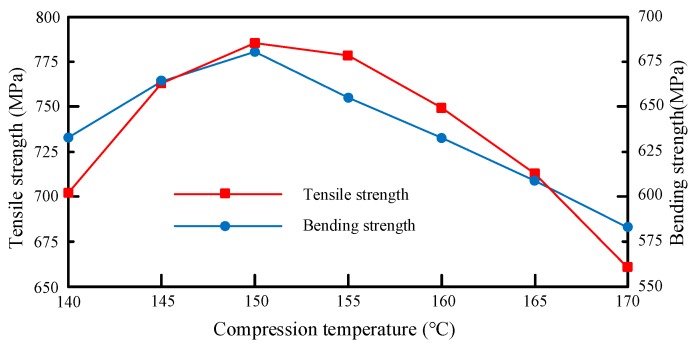
Mapping relationship between compression temperature and mechanical properties.

**Figure 9 materials-12-02430-f009:**
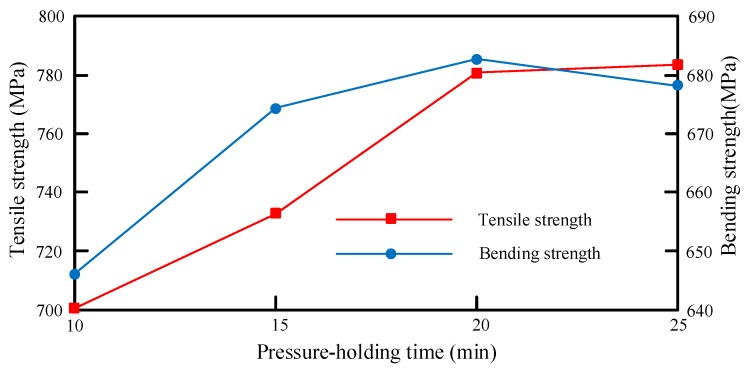
Mapping relationship between pressure-holding time and mechanical properties.

**Figure 10 materials-12-02430-f010:**
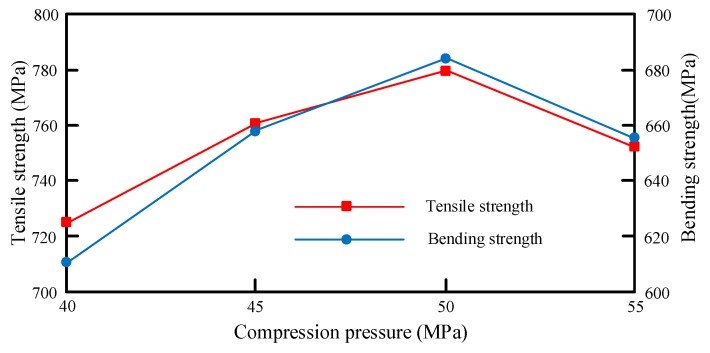
Mapping relationship between compression pressure and mechanical properties.

**Figure 11 materials-12-02430-f011:**
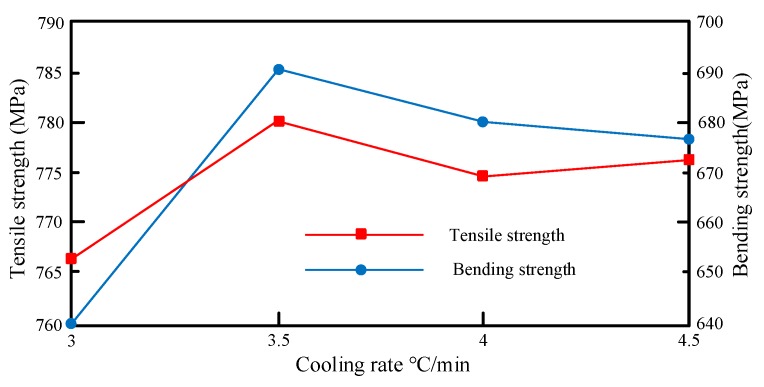
Mapping relationship between cooling rate and mechanical properties.

**Table 1 materials-12-02430-t001:** The characteristic parameters of the prepreg.

Index	Value
Form	12 K; 2 × 2 Twill
Width (mm)	1,000 ± 10
Fabric surface density (g/m^2^)	400 ± 10
Prepreg surface density (g/m^2^)	727 ± 28
Resin content (%)	40 ± 2
Prepreg thickness (mm)	0.44 ± 0.02

**Table 2 materials-12-02430-t002:** The levels of the five parameters in the orthogonal tests.

Level	A	B	C	D	E
	Compression Temperature (°C)	Mold-Opening Temperature (°C)	Cooling Rate (°C/min)	Compression Pressure (MPa)	Pressure-Holding Time (min)
1	140	70	3	40	10
2	150	80	3.5	45	15
3	160	90	4	50	20
4	170	100	4.5	55	25

**Table 3 materials-12-02430-t003:** Orthogonal test schedule.

Test No.	Influencing Factors				
	A	B	C	D	E
1	140	70	3	40	10
2	140	80	3.5	45	15
3	140	90	4	50	20
4	140	100	4.5	55	25
5	150	80	4	55	10
6	150	90	4.5	40	15
7	150	100	3	45	20
8	150	70	3.5	50	25
9	160	90	3	45	25
10	160	100	3.5	50	10
11	160	70	4	55	15
12	160	80	4.5	40	20
13	170	100	4	40	15
14	170	70	4.5	45	20
15	170	80	3	50	25
16	170	90	3.5	55	10

**Table 4 materials-12-02430-t004:** The results of L_16_(4^5^) orthogonal tests.

Test No.	Tensile Strength		Bending Strength		ILSS	
	Test Results (MPa)	Error	Test Results (MPa)	Error	Test Results (MPa)	Error
1	742.16	11.05	627.93	10.15	50.57	3.96
2	753.45	12.47	615.56	9.87	58.71	4.13
3	712.26	11.46	638.37	9.62	54.35	4.64
4	724.13	12.78	620.03	10.58	62.19	4.29
5	698.72	14.21	668.40	11.02	57.08	3.89
6	733.38	13.52	642.66	10.84	72.74	3.78
7	719.75	12.89	628.41	9.81	60.48	5.06
8	688.96	13.46	557.83	15.42	66.36	5.22
9	661.37	10.99	604.07	10.35	59.94	5.48
10	692.62	11.47	626.93	12.77	51.23	3.72
11	723.58	14.23	647.45	12.93	60.90	3.90
12	722.21	13.74	585.28	9.84	56.65	4.27
13	672.49	11.67	572.74	9.21	61.97	4.02
14	663.85	12.10	590.26	10.86	64.04	4.45
15	658.27	12.79	601.82	10.93	63.29	4.67
16	704.32	13.62	610.19	10.83	65.22	3.92

ILSS—interlaminar shear strength.

**Table 5 materials-12-02430-t005:** The results on tensile strengths of the orthogonal tests.

Tensile Strength (MPa)	Influencing Factors				
	A	B	C	D	E
$\delta_{1}$	2932.00	2818.55	2781.55	2870.25	2837.82
$\delta_{2}$	2840.81	2832.65	2839.35	2798.42	2882.90
$\delta_{3}$	2799.78	2811.33	2807.05	2752.11	2818.07
$\delta_{4}$	2698.93	2808.99	2843.57	2850.75	2732.73
$\bar{\delta_{1}}$	733.00	704.64	695.39	717.56	709.46
$\bar{\delta_{2}}$	710.20	708.16	709.84	699.61	720.73
$\bar{\delta_{3}}$	699.95	702.83	701.76	688.03	704.52
$\bar{\delta_{4}}$	674.73	702.24	710.89	712.69	683.18

**Table 6 materials-12-02430-t006:** The ranges of mechanical performance in the orthogonal tests.

Mechanical Performance	A	B	C	D	E	Ranking
Tensile strength (MPa)	58.27	5.92	15.51	29.53	37.54	A > E > D > C > B
Bending strength (MPa)	31.72	17.96	29.11	30.28	37.43	E > A > D > C > B
ILSS (MPa)	7.71	4.13	5.34	2.54	7.56	A > E > C > B > D

**Table 7 materials-12-02430-t007:** Effects of compression temperature on mechanical performance.

**Compression Temperature (**°C**)**	140	145	150	155	160	165	170
**Tensile Strength (MPa)**	701.74	762.90	785.28	778.42	749.16	712.55	660.43
**Bending Strength (MPa)**	632.62	663.81	680.36	654.74	632.53	608.72	582.67

**Table 8 materials-12-02430-t008:** Effects of pressure-holding time on mechanical performance.

**Pressure-Holding Time (min)**	10	15	20	25
**Tensile Strength (MPa)**	712.39	768.90	785.28	776.64
**Bending Strength (MPa)**	640.27	656.43	680.36	681.76

**Table 9 materials-12-02430-t009:** Effects of compression pressure on mechanical performance.

**Compression Pressure (MPa)**	40	45	50	55
**Tensile Strength (MPa)**	710.39	758.91	785.28	755.35
**Bending Strength (MPa)**	624.83	661.32	680.36	652.47

**Table 10 materials-12-02430-t010:** Effects of cooling rate on mechanical performance.

**Cooling Rate (°C/min)**	3	3.5	4	4.5
**Tensile Strength (MPa)**	760.14	785.28	780.17	778.42
**Bending Strength (MPa)**	653.37	680.36	669.45	672.60

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
