# Peer review of "Process Optimization for Compression Molding of Carbon Fiber–Reinforced Thermosetting Polymer"

_materials, 2019, doi:10.3390/ma12152430_

Round 1
Reviewer 1 Report
General Comments
This paper suffers of several shortcomings.
Specific comments
Introduction section is adequate in length but needs to improve in contents about different statistical methods used in materials analysis and explain why has been choose orthogonal and single factor test in front of Taguchi or others. Also the authors need references about used statistical methods.
Results and …….? section
Subsection 3.3 “Results of single Factor Test” needs deeper clarification/ description.
First of all the authors need to explain better how works S-F Test.
I don’t understand why ILSS has lower affectation by the parameters of compression moulding and in S-F analysis it has been rejected (discarded).
When authors has been analysed the effects of different parameters on mechanical performance they use as a constant the optimum parameters (20min, 50T, 3,5ºC/min and 80ºC) that will find it after, why?
Discussion is short and not explain the results only show the results obtained using the “optimal” parameters.
In order to corroborate the best mechanical properties obtained by optimal parameters I think that another analysis technique should be used (i.e SEM).
Author Response
Point 1: Introduction section is adequate in length but needs to improve in contents about different statistical methods used in materials analysis and explain why has been choose orthogonal and single factor test in front of Taguchi or others. Also the authors need references about used statistical methods.
Response 1: In the introduction, the author supplements the contents of different statistical methods by reading relevant references, and explains the reasons for choosing orthogonal experiments and single factor experiments.
Point 2: Results and …….?
Response 2: 3 Result and Discussion, And delete 4.Discussion.
Point 3: Subsection 3.3 “Results of single Factor Test” needs deeper clarification/ description..
Response 3: The author adds deeper clarification and supplements in Subsection 3.3 “Results of single Factor Test”.
Point 4: First of all the authors need to explain better how works S-F Test.
Response 4: The author adds and explain better how works S-F Test in 3.3. Results of Single-Factor Tests.
Point 5: I don’t understand why ILSS has lower affectation by the parameters of compression moulding and in S-F analysis it has been rejected (discarded).
Response 5: It can be seen from the orthogonal test that the change in the range of each parameter has a low influence on the ILSS, so it is omitted in the S-F. In fact, the experiment in this part has not been neglected. After the test, it is found that the results of the changes of various factors have little difference or even crossover. Therefore, the impact of ILSS on the paper is not discussed.
Point 6: When authors has been analysed the effects of different parameters on mechanical performance they use as a constant the optimum parameters (20min, 50T, 3,5ºC/min and 80ºC) that will find it after, why?
Response 6: The best process parameters obtained are the optimum process parameters under the experimental conditions. Due to the limitation of the experimental cost, the discussion cannot be further refined. In the subsequent finite element analysis experiments, further optimization and experimentation will be carried out on the basis of this range. Therefore, the optimal process parameters in the paper will lay the foundation for the subsequent research, but it is relatively optimal.
Point 7: Discussion is short and not explain the results only show the results obtained using the “optimal” parameters.
Response7: In order to better explain the results obtained by the “optimal” parameters, the results are combined with the discussion and more discussed.
Point 8: In order to corroborate the best mechanical properties obtained by optimal parameters I think that another analysis technique should be used (i.e SEM).
Response 8: The sample after the test was scanned by SEM, and the mechanism of action was revealed by analysis, which can more clearly explain the influence of each parameter on the mechanical properties. In this paper, the main influencing factors and effects are analyzed to lay foundation for further analysis of the mechanism of action. At present, SEM carbon fiber prepreg, compression molding and post-experiment carbon fiber experiments are being carried out, and specific mechanism analysis will be carried out in the next paper.

Reviewer 2 Report
this paper prepares PAN-based CFRTC laminated board through compression molding, and carries out orthogonal tests and single factor tests to disclose the effects of different process parameters on the mechanical performance of the CFRTC workpieces. it is an interesting paper and it is advised to be accepted after revisions:
the standard for sample size or cutting criterion should be provided
line 116, results and .... please check it.
in table 4. errors are suggested to be added.
in table 7, 8..9 errors are suggested to be added.
5. surface morpohology of fracture surface are suggested to be given.
the following papers are beneficial to this manuscript:Polymer 143 (2018): 1-9.Composites Part A: 109 (2018): 498-506.
Author Response
Point 1: the standard for sample size or cutting criterion should be provide
Response 1: the standard for sample size or cutting criterion are added in paper in 2.4 Characterization and testing.
Point 2: line 116, results and .... please check it.
Response 2: 3 Result and Discussion, And delete 4.Discussion.
Point 3: in table 4. errors are suggested to be added.
Response 2: In the text, each time the Compression molding sample size is 495*495MM, each sample can be cut into multiple test samples. In the paper, 20 samples are taken each time for experiment and averaged to get the final parameters. The supplements the description in2.4. Result of orthogonal test. In addition, the paper adds standard deviation as an error, In Table 4.
Point 4: in table 7, 8,9 errors are suggested to be added.
Response 2: Table 7,8,9 errors as same as table 4, added in tables. And tables are charts to reflect trends better.
Point 5: surface morpohology of fracture surface are suggested to be given.
Response 2: When the experiments were carried out in the paper, they were all macroscopic experiments, and no SEM microscopic experiments were carried out, so the surface morphology of the fracture surface was not determined.
Point 6: the following papers are beneficial to this manuscript: Polymer 143 (2018): 1-9. Composites Part A: 109 (2018): 498-506.
Response 2: The author has studied the paper seriously and gained a lot of benefits. The article is listed as a reference。

Reviewer 3 Report
In this paper, the authors report a process optimization for compression molding of Carbon fiber reinforced thermosetting composite. Carbon composites are characterized by various mechnical test. Very good results of optimization process such as tensile test are reported.
The application is interesting. The results can be relevant from an application viewpoint.
However, in the current form, the study appears rather superficial. The authors should provide better evidence of their results and conclusions, and try to make a better design of their experiment and deeper analysis of their data.
The paper has several presentation and technical issues that should be addressed before publication.
Here are some suggestions and concerns:
1.Please show and explain the properties such as inner (crystality) and outter information (diameter, length, density, etc...) of carbon nanofibers by SEM image or raman charteristic.
The properties of raw carbon nanofiber is important information for reader.
2.The mechanical testing system was done at a their system, please add the schematic diagram and camera picture image of testing systems for reader.
Author Response
Point 1: Please show and explain the properties such as inner (crystality) and outter information (diameter, length, density, etc...) of carbon nanofibers by SEM image or raman charteristic. The properties of raw carbon nanofiber is important information for reader.
Response 1: The properties of raw carbon nanofibers are important to the reader. The raw materials used in the paper are prepregs, and the carbon fiber is long fiber of Toray T700. The information has been further refined in 2.1 Raw materials. The prepreg is supplied from Maker-tech. According to the commonly used expression method, the internal and external information are indirectly indicated by parameter in Table 1, such as Fabric surface density and Prepreg surface density.
Point 2: The mechanical testing system was done at a their system, please add the schematic diagram and camera picture image of testing systems for reader.
Response 2: According to the relevant standards, the mechanical tests are tested on an MTS universal material testing machine, and in 2.4 Characterization and testing is described in detail. Because the author considered the lack before the test, the camere picture of the test process was not retained during the experiment, only the experimental results were recorded. The current experiment and the material in the paper are different, and we are regret for the camera picture is temporarily unable to be supplemented.

Reviewer 4 Report
The authors report on the process parameters’ optimization towards compression molding of carbon fiber reinforced thermosetting composites. The paper in general contains some novel results with all the required experiments to support the main paper scientific idea”.
However, the following aspects should be considered.

Author Response
Response to Reviewer 2 Comments
Point 1: The first sentence in Abstract (line 9-10) should be deleted. This is a theoretical aspect and it is not relevant to the content of the abstract.
Response 1: The first sentence in Abstract has been deleted.
Point 2: Some fracture surface morphological results (optical microscopy and/ or SEM) should be included from the tensile tests, flexural tests and ILSS tests.
Response 2: 3 Result and Discussion, And delete 4.Discussion.
Point 3: In the introduction part, apart from the mechanical performance/ unique characteristics of CFRTCs, some other aspects e.g. multifunctional FRTCs with sensing (Composites Part B: Engineering Vol. 169, 2019, 37-44), energy harvesting (https://pubs.rsc.org/en/content/articlelanding/2016/ra/c6ra09800b/unauth#!divAbstract), Raman strain sensing (Composites Science and Technology Volume 165, 8 September 2018, Pages 240-249), enhanced interfacial strength (Carbon Volume 73, July 2014, Pages 310-324; Materials & Design Volume 58, June 2014, Pages 1-11; Composites Communications Volume 3, March 2017, Pages 33-37; Journal of Colloid and Interface Science Volume 487, 1 February 2017, Pages 444-457) etc. should be included.
Response 3: The author has studied the papers seriously and gained a lot of benefits. The articles are listed as references.
Point 4: The authors should make clear the difference between “hot pressing” and “compression molding”.
Response 4: In terms of carbon fiber molding, the molding method used in this paper is compression molding.
Point 5: Is it possible to show some images of the final 500x500 laminates.
Response 5: The final laminates has been showed in paper as supplement, but the better one has been cut to experiments.
Point 6: The authors should show schematically the stacking of the laminates for each mechanical test performed as well as the exact sizes/ sample geometry used for each test. Also, how many samples for each test have been tested to follow statistics.
Response 6: All the information inferred in Point 6 are added in paper in 2.4 Characterization and testing.
Point 7: In the 4. Discussion Part, 3 to 4 paragraphs should be included making clear the optimum parameters found and how they affect the mechanical properties. In this part, some fractography should also be included showing some features like crack length, fiber pull-out in cases of low modulus samples, etc.
Response 7: All the experiments in the paper are macroscopic experiments, no microscopic experiments. The microscopic experiments are currently underway, but not all have been completed.
Point 8: In terms of originality, scientific quality, relevance & contribution to the field, this manuscript is of good level.
Response 8: No response.
Point 9: The authors should try to improve the Abstract, Introduction part, Experiential, and Results and Discussion parts, as it has been indicated.
Response 9: The authors has made a number of changes based on the revised comments.
Point 10: Finally, some techniques and additional experimental results will be nice to be included as it has been mentioned, since the paper will be much more attractive to the reader.
Response 10: The authors has made a number of changes based on the revised comments.

Round 2
Reviewer 1 Report
Paper has been improve over the firts version. Introduccions is more complete with new references that helps to understant the context of the reserch. New pictures about the process improve the methodology and results and discusion has updated.
Reviewer 4 Report
The authors have significantly improved the manuscript and can be now accepted for publication.